# Virulence Factors of Meningitis-Causing Bacteria: Enabling Brain Entry across the Blood–Brain Barrier

**DOI:** 10.3390/ijms20215393

**Published:** 2019-10-29

**Authors:** Rosanna Herold, Horst Schroten, Christian Schwerk

**Affiliations:** Department of Pediatrics, Pediatric Infectious Diseases, Medical Faculty Mannheim, Heidelberg University, 68167 Mannheim, Germany; rosanna.herold@medma.uni-heidelberg.de (R.H.); horst.schroten@umm.de (H.S.)

**Keywords:** bacteria, blood–brain barrier, blood–cerebrospinal fluid barrier, meningitis, virulence factor

## Abstract

Infections of the central nervous system (CNS) are still a major cause of morbidity and mortality worldwide. Traversal of the barriers protecting the brain by pathogens is a prerequisite for the development of meningitis. Bacteria have developed a variety of different strategies to cross these barriers and reach the CNS. To this end, they use a variety of different virulence factors that enable them to attach to and traverse these barriers. These virulence factors mediate adhesion to and invasion into host cells, intracellular survival, induction of host cell signaling and inflammatory response, and affect barrier function. While some of these mechanisms differ, others are shared by multiple pathogens. Further understanding of these processes, with special emphasis on the difference between the blood–brain barrier and the blood–cerebrospinal fluid barrier, as well as virulence factors used by the pathogens, is still needed.

## 1. Introduction

Bacterial meningitis, as are bacterial encephalitis and meningoencephalitis, is an inflammatory disease of the central nervous system (CNS). It can be diagnosed by the presence of bacteria in the CNS. Despite advances in treatment, it is still a disease of global dimension, which can end fatally or leave long-term neurological sequelae in survivors [1].

The brain is well protected from invading pathogens by cellular barriers, with the two major barriers being the blood–brain barrier (BBB) and the blood–cerebrospinal fluid barrier (BCSFB) [2,3]. To reach the CNS, blood-borne bacteria must interact with and cross these barriers of the brain. During the course of infection attachment to the host cells is initiated, followed by the hijacking of several host cell pathways by the pathogens. This process can be used by the bacteria to enter host cells with subsequent intracellular survival and, for some pathogens, multiplication. Finally, after crossing into the brain, the bacteria elicit an immune response from the cells within the CNS that might contribute to the inflammatory events leading to disease, with potential disruption of barrier integrity [4].

The traversal or breach of these barriers by meningitis-causing pathogens is defined by a complex interplay between host cells and the pathogens, which use an array of virulence factors to facilitate this interaction, resulting in high morbidity and mortality [1]. These virulence factors are involved during protection in the bloodstream, such as the capsule of both gram-positive and gram-negative bacteria, mediate adhesion to and invasion into host cells, and are responsible for intracellular survival.

## 2. Barriers of the Central Nervous System

The barriers protecting the brain are essential for its function and the stability of its internal microenvironment [5]. Important morphological components found in most of these barriers are specialized intercellular tight junctions between the cells, which are the basis for the barrier property. Various transporters, such as ATP-binding cassette (ABC) transporters, as well as efflux pumps control the movement of molecules across these interfaces, making them another component of barrier function [6]. These mechanisms prevent substances such as toxins, pharmacologically active agents and pathogens from accessing the CNS [4,7,8].

### 2.1. Blood–Brain Barrier

The BBB is formed by brain microvascular endothelial cells (BMECs), astrocytes and pericytes. It presents a structural and functional barrier and acts as an interface between the CNS and the peripheral circulation [2,9]. It has an additional protective function, restricting the free movement of substances across and inhibiting the entry of pathogens and toxins into the CNS. By regulating the passage of molecules, it is furthermore responsible for maintaining the CNS homeostasis. 

The structure of the BBB is defined by BMECs covering the inner surface of the capillaries by forming a continuous sheet of cells interconnected by tight junctions (TJs) [10]. Specific TJ proteins, including occludin and, at the BBB, claudin-5, and claudin-12, but not claudin-3, are responsible for limiting the paracellular diffusion of ion and solutes across the barrier [11,12]. BMECs are characterized by less cytoplasmic vesicles, a larger number of mitochondria and a high amount of intercellular junctions, resulting in a high transendothelial resistance and low paracellular flux [13]. Transporter systems of the BBB include solute carrier-mediated transport, receptor-mediated transport, as well as active efflux and ion transport [14]. The BMECs are supported in shaping the BBB by associated cells such as astrocytes and pericytes [10]. The mechanisms of microbial CNS invasion have been elucidated by the use of in vivo and in vitro studies. To model infections of the CNS, mice and rats are most commonly used. To model the BBB in in vitro cell culture systems, primary human, rodent, and bovine brain microvascular endothelial cells have been used. Extensive studies of the human BBB have been facilitated by the availability of immortalized brain endothelial cells such as human brain microvascular endothelial cells (HBMECs) and hCMEC/D3 [1].

### 2.2. Blood–Cerebrospinal Fluid Barrier

Another major barrier protecting the brain is the BCSFB, which can be separated into a barrier to the inner CSF in the ventricles at the choroid plexus (CP) and barriers to the outer CSF located at the arachnoidea and blood vessels present in the subarachnoidal space [15]. Importantly, the choroidal BCSFB is essential in the trafficking of immune cells into the CNS [3,16]. The epithelial cells displaying polarity and forming tight junctions, which specifically express claudin-1, -2, -3, and -11, are the morphological correlates of the BCSFB at the CP [3,11,12]. The presence of specific transporter systems, such as ABC transporters, and low pinocytotic activity regulating the crossing of substances are key characteristics of the CP [17,18]. Furthermore, the BCSFB is crucial for the protection of the CNS from pathogens [19].

To cause diseases of the CNS bacterial pathogens need to overcome these barriers [1,17]. During this process, several different steps are influenced by bacterial virulence factors. To analyze the role of the CP during infectious diseases of the CNS, animal models can be utilized. Important tools for studying the BCSFB in vitro are primary models of CP epithelial cells, such as primary porcine CP epithelial cells (PCPEC). Another functional model of the BCSFB is a human CP epithelial papilloma cell line (HIBCPP cells) which displays apical/basolateral polarity and a good barrier function [17].

## 3. Stages during the Pathogenesis of BACTERIAL Meningitis

Bacterial meningitis can be caused by a variety of different gram-positive and gram-negative pathogens. There are mechanisms and steps of disease progression that are similar among them as shown in Figure 1. Colonization of the mucosal surfaces is often the first step of disease progression, followed by different strategies of dissemination into the bloodstream [4]. Once the pathogens have entered the bloodstream and successfully counteracted innate and adaptive immune defenses, they have to translocate across the barriers of the brain to invade the CNS and cause severe inflammation. Attachment to the brain endothelium or the epithelial cells of the choroid plexus and subsequent invasion mark the initial step of translocation into the CNS [4].

### 3.1. Attachment and Invasion

To facilitate adhesion and invasion of the barriers protecting the brain, a threshold level of bacteremia has been shown to be required [20,21,22], which is correlated with the severity of infection and likeliness of developing meningitis [23]. However, direct invasion from neighboring infected tissues can occur as well.

Bacterial adhesion to host cell surfaces is a complex process. It involves multiple adhesion molecules of the pathogen interacting with a variety of target receptors. These interactions, which can involve several adhesins of one microbe, can occur in a sequential manner. Hereby, the initial interactions can trigger the expression of further host receptors, which are then targeted by other bacterial adhesins [24].

Many pathogens have been shown to bind to extracellular matrix proteins to facilitate initial attachment to the host cells. Furthermore, binding of bacterial adhesins to specific host cell receptors can in turn induce different signal transduction pathways, resulting in tight attachment or internalization of the bacteria into the host cells [4]. The use of pili or fibrils for invasion of HBMECs was observed for a multitude of meningitis-causing pathogens [8], making them highly important virulence factors for invasion of the CNS.

Endocytosis of pathogens into non-phagocytotic cells is initiated by one of two mechanisms: the “zipper” and the “trigger mechanism”. Some pathogens express surface proteins capable of interacting with transmembrane receptors on the hosts cells which are connected to the cytoskeleton. The “zipper mechanism”, in particular, is defined by an interaction of a bacterial ligand and a host-specific membrane receptor initiating signaling events that lead to internalization of the pathogen through endocytosis [25]. The “trigger mechanism” is a micropinocytosis-related process that involves formation of actin-rich membrane ruffles. These are formed by localized changes in actin dynamics and membrane remodeling triggered by delivery of active effectors, which can be injected by the needle-like structure of a type three secretion system (T3SS), into the cytosol of host cells initiating signaling cascades [26,27].

#### 3.1.1. CNS Entry Routes

Meningitis-causing pathogens most commonly cross host barriers in a transcellular or paracellular manner [2]. These processes are associated with protein interactions between pathogens and the host’s cells. Transcellular traversal is characterized by pathogens crossing the barrier cells without evidence of TJ disruption or traversal between cells [2]. This is accomplished by intracellular invasion of the barrier cells and exploitation of signaling pathways. Paracellular traversal, on the other hand, involves penetration of pathogens between the host’s cells and can occur with and without permanent disruption of TJs [2,4]. Furthermore, the release of bacterial toxins can lead to disruption of barrier function and promote paracellular traversal. Another means of entry is the “Trojan-horse” mechanism, which describes penetration of the barrier by transmigration within infected phagocytes [2]. It has been suggested that the infected phagocytes adhere to the luminal side of brain capillaries. This can occur with and without the activation of BMECs and is followed by either transcellular or paracellular traversal of the BBB [28].

#### 3.1.2. Signal-Transduction Mechanisms and Cytoskeletal Rearrangements

Meningitis-causing bacterial pathogens have been shown to utilize host cell signaling molecules to facilitate infection. The mechanisms deployed can vary between the different pathogens, as well as the host tissues.

To both enter and leave the host cells, several pathogens have developed mechanisms to use the actin polymerization machinery of the host [29,30]. Pathogens not only use this process to spread throughout the cells, but have also developed mechanisms to subvert these regulatory mechanisms [31]. Furthermore, meningitis-causing pathogens use different signal-transduction mechanisms that result in rearrangements of the actin cytoskeleton [2]. This allows the pathogens to initiate attachment and entry of host cells, movement within and among cells as well as vacuole formation. This complex process of actin cytoskeletal remodeling can involve many factors, such as Rho-family GTPases and a variety of actin-binding proteins [30,32].

The innate immune system of the host can be triggered by various molecules which are characteristic for the bacteria. These pathogen-associated molecular patterns (PAMPs) are then recognized by eukaryotic pattern recognition receptors (PRRs), which in turn induce signaling cascades such as the nuclear factor κB (NF-κB) and mitogen-activated protein kinase (MAPK) pathways [33,34]. Activation of these signaling cascades triggers pro-inflammatory responses like the up-regulation of cytokines [35].

### 3.2. Intracellular Survival

#### 3.2.1. Multiplication and Intracellular Survival

Extracellular multiplication is the most common process for propagation by pathogenic bacteria. Furthermore, replication and persistence inside host cells has been demonstrated for a variety of meningitis-causing pathogens as well [36]. Entry into the host´s cells and replication within these protects the pathogens from clearance by the complement system and circulating antibodies. However, these pathogens have to overcome several cellular defense mechanisms such as the upregulation and secretion of neutrophil-specific factors in HBMECs. This response by the BBB is assumed to serve to recognize pathogens resulting in their clearance. However, overactivation of the cellular response through continued exposure to the pathogens could result in increased inflammation and compromised barrier integrity [8]. 

#### 3.2.2. Disruption of Barrier Integrity and Inflammatory Response

Characteristically, bacterial meningitis is accompanied by a severe inflammatory response leading to neuronal damage. Activation of the transcription factor NF-κB upon invasion of the brain barrier tissues results in high levels of inflammatory cytokines in the blood and CSF [37]. This proinflammatory response can be triggered by bacterial cell wall components. Examples would include the lipopolysaccharide (LPS) for gram-negative bacteria, and lipoteichoic acid (LTA) for gram-positive bacteria. Furthermore, increased permeability of the barriers can be triggered by both bacterial toxins, as well as the initiation of host inflammatory pathways in response to the infection [38]. Tissue damage during bacterial meningitis arises from the initiated inflammatory cascade involving cytokines and chemokines, as well as proteolytic enzymes and oxidants [4]. Consequences of the release of these inflammatory substances, caused by multiplication of pathogens in the CNS, are damage of neurons and edema [39]. The hosts’ immune response is therefore unable to embank infection of the CNS, and may even contribute to adverse events during bacterial meningitis [4].

The release of proinflammatory molecules not only increases permeability of the BBB but also attracts leukocytes to the CNS [40]. Cell death can be caused by cytokines, reactive oxygen species, reactive nitrogen species and matrix metalloproteinases [41,42,43,44,45].

## 4. Roles of Bacterial Virulence Factors During Invasion Through the Barriers of the CNS

To be able to enter the CNS, bacterial pathogens use several virulence factors, which are involved in the different steps of pathogenesis. Of major importance is the capsule of both gram-positive and gram-negative bacteria. The capsular polysaccharide has a protective function in bloodstream survival [46]. However, it was observed to attenuate invasion of the BBB and BCSFB [47,48,49]. This could result from electrostatic repulsion or from the masking of bacterial surface structures [8]. The necessity of the capsule for the pathogens survival in the blood but simultaneous hindrance of invasion of the host’s tissues indicates a need for the regulation of capsule expression [50].

Besides the capsule, a multitude of further virulence factors are used by gram-positive and gram-negative bacteria [51]. These include, among others, adhesins and internalins, pore-forming toxins, and factors involved in intracellular movement as well as cell-to cell spread and are summarized in Figure 2.

In this review, we will focus particularly on virulence factors, for which evidence for an involvement during brain entry across the blood–brain barriers has been proposed. A summary of these virulence factors for gram-positive and gram-negative bacteria is given in Table 1 at the end of this section.

### 4.1. Gram-Positive Bacteria

#### 4.1.1. Listeria Monocytogenes

*Listeria monocytogenes* (*L. monocytogenes*) is a facultative intracellular gram-positive bacterium. It can traverse several physiological barriers and finally enter the brain via the BBB or BCSFB, especially in immunocompromised individuals [52]. It is ingested through highly contaminated food by the host. Once ingested, *L. monocytogenes* traverses the intestinal epithelial barrier into the lamina propria followed by dissemination of the pathogen via the lymph and blood [53]. *L. monocytogenes* has multiple target organs, including the liver and spleen, and can enter the CNS across the barriers of the brain [53]. In addition to direct traversal of the BBB and BCSFB via the transcellular route, transportation across the BBB within leukocytes and retrograde migration within axons of cranial nerves have been described [54,55].

*L. monocytogenes* can enter non-phagocytotic cells by hijacking the host’s receptor-mediated endocytosis machinery using the zipper mechanism. The two major invasion proteins of *L. monocytogenes* are internalin (InlA) and InlB, which bind to eukaryotic cell membrane members E-cadherin and tyrosine kinase receptor protein Met, respectively. These interactions induce receptor-mediated endocytosis of the pathogen. *L. monocytogenes* has been demonstrated to use one or both internalins to mediate invasion of the BBB and BCSFB [25,56,57]. A recent study has further demonstrated the importance of the bacterial surface protein InlF, showing that interaction with surface vimentin was required for an optimal colonialization of the brain [58].

The MAPK signaling cascade is activated during the invasion of *L. monocytogenes* [35,59,60]. In a model system of the BCSFB consisting of choroid plexus epithelial cells, the requirement of MAPK activation for listerial entry was demonstrated. Both extracellular signal-regulated kinases (ERK) 1 and 2 and p38 inhibition resulted in decreased bacterial invasion into this model system suggesting their involvement in the pathogens traversal of the BCSFB [34].

It was previously described that ubiquitination of E-cadherin and Met leads to the recruitment of the clathrin-mediated endocytosis machinery. This in turn results in the polymerization of the actin cytoskeleton. During this process, dynamin recruits several factors that result in two waves of actin rearrangements and subsequently result in the entry of the pathogen inside of vacuoles [61,62,63]. Accordingly, an in vitro study using a model of the BCSFB based on HIBCPP cells, revealed that *L. monocytogenes* invasion is inhibited if dynamin-mediated endocytosis is blocked [34].

Another essential virulence factor of *L. monocytogenes* is the pore-forming cytolysin Listeriolysin O (LLO). Activation of the NF-κB signaling pathway by LLO was reported in the human embryonic kidney HEK-293 cell line [64], as well as MAPK signaling [65,66]. It is secreted by *L. monocytogenes* and promotes the pathogens intracellular survival. After entering the host cell, lysis of the vacuole is initiated through LLO and the bacterial phospholipases PlcA and PlcB, and followed by intracellular spread in the cytoplasm [61].

Once *L. monocytogenes* has reached the cytoplasm of the host´s cells, it has been demonstrated to move around and enter neighboring cells using actin comet tails and membrane protrusions to facilitate its spread [61,67]. This F-actin-based intracellular motility is dependent on the expression of another essential listerial virulence factor, ActA [68].

Activation of the NF-κB signaling pathway is, as previously described, achieved through LLO. Another mechanism involving NF-κB is its activation by InlC, which is secreted intracellularly. It can directly interact with the subunit of the IκB kinase complex, IKKα. By phosphorylating IκB, this complex is critical for the activation of NF-κB, a major regulator of innate immune response. InlC was shown to impair phosphorylation of IκB, thereby scaling down the hosts immune response [69], and is also involved in cell-to-cell spread [70]. 

#### 4.1.2. *Streptococcus suis*

*Streptococcus suis* (*S. suis*) is a zoonotic gram-positive bacterium and one of the most important porcine bacterial pathogens. Serotype 2 of *S. suis* has been described to be a major cause of meningitis, especially in South and East Asia [71]. To reach the CNS, *S. suis* has to colonize the host and traverse epithelial barriers in order to reach the bloodstream, where it needs to survive. *S. suis* has been demonstrated to cross the BBB and the BCFSB in human in vitro models as well as in porcine models [48,71,72,73].

The presence of a capsule is essential for survival in the bloodstream. However, it was demonstrated to attenuate invasion for *S. suis* in epithelial cells [48,72,74]. A link between capsule expression and carbohydrate metabolism has been described, indicating adaptation of *S. suis* to different environments. High concentrations of nutrients, as found in the bloodstream, coincided with high expression of the capsule, whereas in the CNS, which is low in nutrients, expression was reduced [50,75]. Attachment of *S. suis* to BMECs has been demonstrated in human and porcine in vitro models of the BBB [76,77]. Invasion has been reported in porcine models but at very low rates [73,78]. During the adhesion process, in these in vitro model systems, the capsule had no effect on adherence [76]. In contrast, in porcine and human models of the BCSFB, both attachment and significant invasion of *S. suis* strains were demonstrated. The use of unencapsulated mutants further increased invasion rates, indicating a role of the capsule and regulation of its expression [48,72]. Other important virulence factors for invasion of the barriers of the brain are cell wall components such as lipoteichoic acid, LPXTG-anchored proteins as well as lipoproteins [79].

Also, enolase has been identified as a virulence factor of *S. suis.* Previously thought to act only as a glycolytic enzyme, this protein, with a highly conserved sequence, has been implicated in the invasion process of various pathogens. For *S. suis*, enolase has been shown to increase BBB permeability as well as promoting the release of interleukin IL-8 [80]. Transmigration might further be promoted by the thiol-activated cytolysin suilysin, which induces pore formation in membranes. Furthermore, the use of bacterial mutants lacking suilysin has shown that it is not essential for invasion of the host [81]. It has, however, been demonstrated to promote association with epithelial cells, making it another major virulence factor of the pathogen [82].

The upregulation of different cytokines and chemokines in response to *S. suis* infection in BMECs has been reported. Exemplary is the induction of IL-6 and IL-8, stimulated by *S. suis* [83]. An inflammatory response was also described in a porcine model of the BCSFB. Here, induction of tumor necrosis factor (TNF) α and matrix metalloproteinase (MMP)-3 gene expression were described. This was paralleled by rearrangements of the tight junction proteins ZO-1, occludin and claudin-1, and loss of actin at the apical cell pole as well as the induction of stress fiber formation at the basolateral side of the barrier [84]. The expression of TNFα after stimulation by *S. suis* in porcine choroid plexus epithelial cells further promoted adhesion and transmigration of polymorphonuclear neutrophils (PMN) through the barrier, which is a critical step during bacterial meningitis. Interestingly, some PMNs contained internalized *S. suis*, indicating the possibility of the pathogen exploiting the Trojan-horse mechanism [85].

#### 4.1.3. *Streptococcus Pneumoniae*

*Streptococcus pneumoniae* (*S. pneumoniae*) is a gram-positive pathogen which is the major cause of bacterial meningitis in the developing world [86]. Close to 30% of individuals carry *S. pneumoniae* asymptomatically. Nasopharyngeal colonization is followed by systemic invasion and access to the bloodstream. Invasion of the CNS via the barriers of the brain is the major cause for meningitis [87]. Furthermore, olfactory neuron invasion was observed to be an entry route for *S. pneumoniae* [88]. To facilitate invasion of the CNS, *S. pneumoniae* utilizes several virulence factors such as the pneumococcal capsule and surface proteins as well as secreted proteins.

Bacterial interactions with the host and the subsequent development of bacterial meningitis are promoted by a high level of bacteremia [4]. Accordingly, attachment of *S. pneumoniae* to the choroid plexus in an in vivo mouse model was observed only during late stages of infection with high levels of bacteremia [87]. Further studies of the interaction with the BCSFB would be needed to determine if *S. pneumoniae* can use it as entry gate to the CNS.

The capacity of *S. pneumoniae* to invade host tissues is majorly determined by its capsule. Similar to other pathogens capable of causing bacterial meningitis, survival in the bloodstream is dependent on maximum capsule expression [89,90], whereas attachment to host tissues is hindered by its presence [91]. As the binding of various pneumococcal surface proteins is hindered by the capsule, altered expression of the capsule through quorum sensing and phase variation has been described [92,93].

Pneumococci have been demonstrated to use multiple virulence factors to initiate attachment to HBMECs in several in vitro studies. The platelet endothelial cell adhesion molecule (PECAM-1) has been implicated as receptor for neuraminidase A (Nad A) in pneumococcal attachment [94,95]. Importantly, NadA has been described as an important virulence factor, anchored in the cell wall of *S. pneumoniae* that can cleave sialic acid of the host substrates. It has been implicated in triggering the transforming growth factor-β (TGF-β) signaling cascade during the interaction with the BBB, resulting in decreased barrier integrity and an increase in invasion [96]. Another important receptor involved in pneumococcal BBB invasion is the polymeric immunoglobulin receptor (pIgR) as the major adhesin of the pneumococcal pilus-1, RrgA, was observed to bind to it as well as PECAM-1. Human platelet-activating factor receptor (PAFR) was demonstrated to be further involved in attachment, together with the before mentioned PECAM-1 and pIgR [97]. Additionally, a second type of pneumococcal pili has been described to mediate adhesion to host cells [98]. Furthermore, choline binding protein PspC was shown to bind only to pIgR [97], and interaction with the laminin receptor by *S. pneumoniae* is initiated by choline-binding protein A (CbpA) [99]. A proteome-based approach in a mouse meningitis model, addressing adaptive capabilities of the pathogens to a defined host compartment, has highlighted a crucial role for two highly expressed pneumococcal proteins; ComDE, a regulatory two-component system, and AliB, a substrate-binding protein of an oligopeptide transporter, in pneumococcal meningitis [100].

A major virulence factor of *S. pneumoniae* is pneumolysin, a pore-forming toxin which acts in a cholesterol-dependent manner. Pneumolysin has recently been demonstrated to induce the expression of CERB-binding protein (CBP), a coactivator of transcription. This results in the release of TNF-α and IL-6, which in turn lead to increased permeability of the BBB through increased apoptosis of the cells both in vivo and in vitro [101]. It has furthermore been implicated in paracellular traversal of the BBB by *S. pneumoniae* as a result of reduced barrier integrity [102].

In addition to the disruption of the BBB by pneumolysin, generation of H_2_O_2_ through α-glycerophosphate oxidase (GlpO) was observed to have a cytotoxic effect on HBMECs [103]. A study using rat brain tissues investigated the effect of *S. pneumoniae* infection on nucleotide-binding oligomerization domain 2 (NOD2) and inflammatory factors, suggesting that NOD2 may hold a role in the activation of inflammatory pathways and subsequent BBB damage [104].

A study by Coutinho et al. analyzed the CSF of patients with pneumococcal meningitis, detecting high levels of pro-inflammatory cytokines such as TNF-α, IL-1β, and IL-6 and anti-inflammatory cytokines IL-10 and TGF-β. Furthermore, chemokines IL-8, MIP-1a and MCP-1 were detected [45]. Furthermore, an in vivo study in neonatal Wistar rats demonstrated an increase in cytokines prior to BBB breakdown after induction of pneumococcal meningitis [105]. Entry of the CNS by the pneumococci is followed by rapid multiplication. Components released by the pathogen during this process are then recognized by PPRs resulting in a strong inflammatory response and subsequent BBB impairment [4].

#### 4.1.4. Group B Streptococcus

*Group B streptococcus* (GBS, *Streptococcus agalactiae*) is a β-haemolytic, gram-positive pathogen and the leading cause of meningitis in human neonates [106]. Classification of GBS strains is done by sequence type based on an allelic profile of seven loci [107]. The development of GBS meningitis is dependent on bloodstream survival and development of a high level of bacteremia of the pathogen, followed by the disruption of the BBB or possibly the BCSFB. This is followed by the multiplication of GBS in the CNS, culminating in severe inflammation and neural damage [4]. The necessity of high-level bacteremia indicates that bloodstream survival is an important virulence factor of the pathogen along with the sialylated GBS capsular polysaccharide [108]. Additionally, reduction of capsule expression by GBS was demonstrated to increase virulence and intracellular persistence [49]

Direct interaction of GBS with the BBB endothelium has been demonstrated, resulting in traversal of the barrier and subsequent infection of the CNS [49,109]. Both direct invasion of the BBB and/or brain invasion as a direct result of increased permeability of the barrier have been observed [4]. The use of in vitro models has further shown transcellular crossing of the pathogen [109]. To this end, a variety of virulence factors have been described. One of these factors is a surface anchored novel protein specific for the GBS ST-17 clone, which is associated with meningitis in infants after the first week of life, called hypervirulent GBS adhesin (HvgA) and is required for GBS hypervirulence. Increased adherence of strains expressing HvgA was detected for intestinal epithelial cells, choroid plexus epithelial cells, and microvascular endothelial cells of the BBB [110]. Another necessary determinant of the GBS interaction with the BBB is the expression of cell-wall anchored pili [111]. Interestingly, these pili displayed similar function in adhesion and invasion of GBS, given that the role of pili has been best described for gram-negative bacteria. Two proteins involved in the formation of the pili, encoded by *pilA* and *pilB,* were shown to facilitate the interaction with the BBB, wherein PilA is promoting attachment of GBS and PilB is mediating internalization of the bacterium [112]. Interaction of PilA with collagen promotes its interaction with the α_2_β_1_ integrin, initiating the integrin signaling machinery [38]. Interaction with HBMECs is further enabled by the GBS fibronectin-binding protein streptococcal fibronectin binding protein A (SfbA) [113]. Additionally, GBS serine-rich repeat (Srr) glycoprotein was suggested to both promote bloodstream survival, facilitated by the adherence to fibrinogen, as well as adhesion to HBMECs [114]. An antigen I/II family adhesin, BspC, was recently demonstrated to interact with host cell vimentin during the pathogenesis of GBS meningitis, thereby promoting adherence of the pathogen in vitro as well as contributing to the development of GBS meningitis in vivo [115]. 

During invasion of type III GBS in HBMECs, tyrosine phosphorylation of the focal adhesion kinase (FAK) was demonstrated. Not only was the phosphorylation of the FAK required for invasion of GBS, it further induced association with PI3-kinase and paxillin, an actin filament adaptor protein [116]. BBB penetration of GBS was shown to also involve the *invasion associated gene A* (*iagA*). Mice challenged with a mutant version of this gene, encoding a glycosyltransferase homolog, developed bacteremia comparable to the wild type (WT) mice but had significantly lower mortality. In addition, the *IagA* gene encodes an enzyme, the glycolipid diglucosyldiacylglycerol, which functions as a cell membrane anchor for LTA, indicating that proper LTA anchoring is necessary for invasion of GBS into the BBB [117]. 

Similarly to *S. pneumoniae*, GBS can secrete a pore-forming toxin to disrupt barrier function in infected BMECs [109]. In addition, a further consequence of the interaction of PilA with α_2_β_1_ integrin is the activation of host chemokine expression as well as neutrophil recruitment, which was correlated with increased permeability of the BBB [38]. Furthermore, GBS hyaluronidase HylB was demonstrated to induce BBB opening in a dose-dependent manner [118]. Another factor influencing BBB dysfunction was described during infection of induced pluripotent stem cell-derived brain endothelial cells with GBS, which resulted in the inhibition of P-glycoprotein, an important efflux transporter for the maintenance of brain homeostasis [119]. Barrier disruption and subsequent bacterial passage was further demonstrated to be promoted by reduced expression of tight junction components ZO-1, Claudin-5 and Occludin in HBMECs. This was facilitated by the induction of transcriptional repressor Snail1 and sufficient to promote tight junction disruption. This process, which was shown to be dependent on ERK 1/2 MAPK signaling as well as bacterial cell wall components, marks another mechanism of BBB disruption by GBS [120]. 

### 4.2. Gram-Negative Bacteria

#### 4.2.1. *Escherichia coli*

*E. coli,* a gram-negative bacillary organism, is a common cause of meningitis and still an important cause of mortality and morbidity throughout the world. Circulating *E. coli* have been shown to traverse the BBB and the BCSFB as a result of hematogenous spread [22,121]. Expression of the K1 capsule and *O*-lipopolysaccharide are critical determinants of *E. coli* meningitis, especially in neonates [122,123,124]. Several factors have been demonstrated to influence *E. coli* invasion of the CNS such as a high level of bacteremia for the invasion of the blood–brain barrier, as well as a variety of virulence factors that initiate binding and translocation into the CNS [2]. 

The first step in *E. coli* invasion of the CNS is attachment to the cells of the brain barriers. The two major virulence factors associated with attachment to the blood–brain barrier are type 1 fimbriae and OmpA [125,126]. The virulence factor IbeA has been associated with the subsequent invasion process [127], as well as cytotoxic necrotizing factor 1 (CNF1) [128]. Deletion of *ompA* and *ibeA* reduced infection of choroid plexus epithelial cells in a human model of the BCSFB, whereas deletion of *fimH* enhanced invasion but simultaneously decreased adhesion of *E. coli* strains in the same model [121]. In an in vitro model of the BBB, Ecgp was identified as receptor for OmpA [129].

To promote internalization, *E. coli* induces rearrangements of the actin cytoskeleton. In HBMECs, this has been demonstrated to trigger a zipper-like mechanism that envelops the bacterium and initiates internalization into the cell. This process is dependent on both the actin cytoskeleton as well as microtubules [4]. Induction of tyrosine phosphorylation of FAK and cytoskeletal proteins by *E. coli* was demonstrated in an in vitro model of brain endothelial cells [130]. In addition, phosphatidylinositol 3-kinase (PI3K) interaction with FAK and PI3-kinase signaling was shown to be necessary for successful invasion of *E. coli* in HBMECs [131]. In turn, PI3K activates phospholipase PLCγ, resulting in increased Ca^2+^ levels in the cells [132]. The cell adhesion molecule ICAM-1 was selectively upregulated during invasion of *E. coli* in brain endothelial cells through the interaction of OmpA and its receptor Ecgp. This upregulation was dependent on the previously described PI3K signaling pathways, as well as protein kinase C (PKC)-α and NK-κB signaling [133].

Traversal of *E. coli* across the BBB has been studied extensively using HBMECs as in vitro models [22]. Transmission electron microscopy revealed the pathogen crossing the HBMECs in membrane-bound vacuoles without intracellular replication [134]. Infection of HBMECs by *E. coli* K1 was further demonstrated to activate caveolin-1, resulting in the uptake of the pathogen via the caveolae. Furthermore, caveolin-1 interacts with phosphorylated protein kinase Cα at the site of *E. coli* attachment [135]. To promote intracellular survival within the vacuoles, the *E. coli* K1 capsule is essential. It can modulate the maturation of the vacuoles and prevent fusion with lysosomes, thereby enabling traversal of live bacteria to the CNS [136]. There are gene clusters, essential for the production of the precursors that are the basis of the K1 capsule, called the *neuDB* genes. They were identified as an essential virulence factor promoting intracellular survival in HBMEC [122,136]. Furthermore, *neuDB* was also shown to be of importance during infection of an in vitro model of the BCSFB [121].

The breakdown of barrier function and neuroinflammation are considered key mechanisms in the invasion of the brain by pathogenic *E. coli.* In brain endothelial cells, upregulation of platelet-derived growth factor-B (PDGF-B) and ICAM-1 was demonstrated after infection with *E. coli* [137]. Further in vivo and in vitro studies suggest an involvement of PDGF-B in BBB permeability, mediating breakdown of tight junction proteins. Upregulation of ICAM-1 on the other hand was shown to initiate the inflammatory response of the CNS, mediating neutrophils or monocyte recruitment during infection [138]. The activation of PKC–α and its association with vascular-endothelial cadherins at the TJs of HBMECs resulted in increased cellular permeability and decrease in transendothelial electrical resistance by releasing β-catenin from the junctions. Notably, only *E. coli* strains expressing *ompA* could induce this increase in barrier permeability [139]. The infection of HBMECs with *E. coli* was further demonstrated to promote the production of nitric oxide (NO) by activating inducible nitric oxide synthase and, as a result, displayed enhanced invasion rates and increased permeability of HBMEC monolayers [140]. NO production was hypothesized to additionally be triggered by the modulation of pterin synthesis, which is involved in cell differentiation, pain modulation as well as mRNA stability. Infection of HBMECs was demonstrated to induce the rate-limiting enzyme in pterin synthesis, guanosine triphosphate cyclohydrolase (GCH1), indicating its role in the invasion process. GCH1 further interacts with Ecgp96, the receptor for OmpA [141]. These findings indicate an essential role for NO during *E. coli* invasion of the BBB. The response of HBMECs to meningitic and non-meningitic *E. coli* infections highlighted the role for macrophage migration inhibitory factor (MIF) during infection. MIF, a proinflammatory cytokine that has been described as a major factor during infection and septic shock, was demonstrated to have a role in BBB damage, as evidenced by the induction of a significant decrease in ZO-1 and occludin, as well as inflammation [142]. 

A further study described the role of *E. coli* K1 virulence factor *cglD* in polymorphonuclear leukocyte transendothelial migration [37]. A follow up analysis demonstrated a contribution of *cglD* to NF-κB pathway activation in HBMECs, promoting PMN adhesion and transendothelial migration across the BBB [143].

#### 4.2.2. *Neisseria Meningitidis*

*Neisseria meningitidis* (*N. meningitidis*) is a human-specific gram-negative bacterium. It can colonize the nasopharynx extracellularly and is often non-pathogenic and commensal. Some strains can cause life-threatening diseases such as meningitis. To reach the barriers protecting the CNS, *N. meningitidis* has to overcome the mucosal epithelium and enter the bloodstream. To survive in the bloodstream and subsequently enter the CNS, *N. meningitidis* utilizes different virulence factors. These protect the bacterium from being killed by the hosts complement system or other effectors. They include the polysaccharide capsule and other surface structures, such as pili and other adhesins [144]. Factor H-binding protein is another important factor for bloodstream survival and evading the host innate immune system by binding factor H, which is a negative regulator of complement activation and alternative pathway and is bound to the surface of the pathogen [145,146]. *N. meningitidis* has been proposed to cross both the BBB and the BCSFB to the inner and the outer CSF [48,147,148,149].

The polysaccharide capsule of *N. meningitidis* is a major contributor to meningococcal disease and has been described as its main virulence factor. It can undergo genetic regulation and has the capability to mask the function of non-pilus adhesins [150]. While the capsule is essential for bloodstream survival, adhesion and invasion of host tissues are inhibited by the capsule [151]. Attenuated invasion was further described for capsulated strains of *N. meningitidis* in an in vitro model of the BCSFB [48]. The loss of the bacterial capsule for members of group B and group C meningococcal strains lead to increased uptake into HBMECs [152], and capsule and pili of *N. meningitidis* are downregulated upon contact with epithelial cells [153].

Adherence of *N. meningitidis* to host cells is facilitated by different virulence factors such as pili and surface exposed proteins like Opa and Opc, which further contribute to meningococcal disease. The type IV pili are a crucial adhesin expressed by *N. meningitidis*. They are involved in the attachment of capsulated virulence strains to host surfaces, extending from the bacterial surface through the capsule [154]. These long filamentous structures, made up of the major pilin protein and three minor pilins, promote adhesion of *N. meningitidis* to human endothelial cells via their interaction with CD147 [155]. In addition to the major pilin, type IV pili adhesion is also dependent on the expression of the outer membrane protein PilC [156,157,158]. Although affinity of pilin monomers to CD147 is weak, their assembly into the type IV pili as well as high expression of CD147 on the BBB supports the interaction [155]. Another group of adhesin-like structures, termed minor adhesion proteins, is expressed at low levels in vitro but may be of importance in in vivo environments which have been shown to alter the transcriptome of *N. meningitidis* [159].

While host, as well as tissue specificity, is determined by the pili and Opa proteins, invasion is facilitated by both Opa and Opc [160]. Opa proteins have been shown to bind to the carcinoembryonic antigen-related cellular adhesion molecule (CEACAM) receptor family, as well as the extracellular matrix proteins fibronectin and/or vitronectin [161,162]. An increase in adhesion and entry into HBMECs has been reported for *N. meningitidis* expressing Opc. This effect is facilitated by binding of Opc to serum vitronectin or fibronectin [163]. Also, activation of the acid sphingomyelidase/ceramide system, which involves clustering of ErbB2, an important receptor involved in bacterial uptake, by Opc-expressing *N. meningitidis* played a major role in determining the pathogens invasiveness [164]. Furthermore, the bacterial adhesins PilQ and PorA were shown to bind to the laminin receptor, thereby initiating contact with the BBB [99].

Attachment and invasion of human cells by three hypervirulent serogroup B strains of *N. meningitidis* is furthermore promoted by the *Neisseria* adhesin A (NadA), a phase-variable, surface-exposed protein [165,166,167]. For its interaction with the host, different membrane proteins have been suggested. NadA was shown to target human β1 integrin subunits [150]. Also, a genome-wide microarray analysis provided evidence for an interaction between NadA and the endothelial low-density oxidized lipoprotein receptor 1 (LOX-1) [168]. Following initial binding to the cell surface, the formation of microcolonies was observed, which in turn promoted the formation of specific molecular complexes called “cortical plaques”. These structures contain accumulated ezrin, moesin, tyrosine-phosphorylated proteins, ICAM-1/-2, CD44 and epidermal growth factor receptors (EGFR) as well as localized polymerized cortical actin, resulting in major reorganization of host cell morphology. Changes in cell surface morphology are a prerequisite for bacterial uptake and the formation of microvilli-like cellular projections which protect the microcolonies from shear stress of the bloodstream [169].

Following attachment, *N. meningitidis* induces host cell signaling events to facilitate invasion into host tissues. These include the recruitment of ezrin as well as the activation of Src kinase and cortactin [170,171,172]. Furthermore, the FAK is essential for integrin-mediated internalization of *N. meningitidis* in HBMECs, enabling endocytosis through the interplay between FAK, Src and cortactin. This is facilitated by *N. meningitidis* using the integrin signaling pathways to mediate signaling from activated integrins upon attachment to the cytoskeleton [173]. These events promote the reorganization of the actin cytoskeleton resulting in the formation of membrane protrusions that take up the pathogens by surrounding them for internalization [144]. Genes involved in cytoskeletal reorganization were shown to be differentially expressed after infection of HBMECs [174]. Analysis of whole-cell lysates of human endothelial and epithelial cells have revealed an interaction of the Opc protein with alpha actinin, a modulator of various signaling pathways and cytoskeletal functions [175].

Disruption of the BBB integrity was observed in in vitro studies as a consequence of opening intracellular junctions [176]. Furthermore, cell detachment is initiated through activation of MMP 8, promoting cleavage of the TJ protein occludin [177]. Alterations of intracellular junctions were observed in a human brain microvascular endothelial cell line, potentially opening up a paracellular route of crossing the BBB into the CNS through recruitment of the Par3/Par6/PKCζ polarity complex [178].

Meningococcal disease is accompanied by an acute inflammatory response [179]. The transcription factor (TF) NF-κB, associated with the release of proinflammatory cytokines and chemokines during the inflammatory response, was shown to be active in an in vitro CP epithelial model after infection with *N. meningitidis*. Its activation was believed to arise from heterodimerization of TLR2 and TLR6 [33]. In HBMEC the p38 MAPK had an impact on the release of IL-6 and IL-8, whereas the c-JUN N-terminal kinases 1 and 2 (JNK1 and JNK2) were important for invasion [180].

#### 4.2.3. *Haemophilus Influenzae*

*Haemophilus influenzae* (*H. influenzae*) is a gram-negative bacterium capable of colonizing the upper respiratory tract. The makeup of the polysaccharide capsule determines the pathogens classification into serotypes a to f, of which *H. influenzae* serotype b (*Hib*) is responsible for the most severe infections, such as meningitis, especially in children under the age of 5 years [181]. Although there is widespread vaccination against *Hib*, some populations remain vulnerable and the vaccines do not protect against other serotypes [182]. Furthermore, in areas where Hib vaccines are used, nontypeable *H. influenzae* (NTHi) strains can now be the most common cause of meningitis [183]. Similar to the before described pathogens, *H. influenzae* has to cross the epithelial barrier of the upper respiratory tract and, after dissemination and survival in the bloodstream, cross the barriers of the brain to enter the CNS [184]. Traversal of both the BBB and BCSFB has been demonstrated for *H. influenzae* [47,51]. 

Attachment to host cells is enabled by the bacterial capsule and fimbriae which are subject to reversible phase variation [185]. *H. influenzae* shares a common strategy to enter endothelial cells with *S. pneumoniae* and *N. meningitidis*, which involves binding to the PAFR, mediated by phosphorylcholine (ChoP) [186,187]. This interaction results in the entry of the pathogens into the BBB through activation of β-arrestin–mediated uptake [188]. Binding of PAFR by lipooligosaccharide (LOS) glycoforms containing ChoP was also demonstrated during invasion NTHi. This binding resulted in the activation of host cell signaling by coupling with pertussis toxin-sensitive (PTX) heterotrimeric G protein complexes and invasion of the pathogen. It was further suggested that this mechanism was more efficient than micropinocytosis [187]. Binding to laminin receptor, another shared mechanism of these pathogens, is initiated by OmpP2, facilitating the interaction with the brain endothelium by *H. influenzae* [99].

In an in vitro model of the BCSFB, *Hib* as well as clinical isolates of *Hib* and *H. influenzae* serotype f (*Hif)* were shown to adhere and invade the HIBCPP cells as intracellular bacteria. Both fimbriae and the capsule lead to attenuated invasion [47]. Also, a study using *H. influenzae* serotype a (*Hia*) and a co-culture of HBMECs and pericytes demonstrated an activation of stimulated A_2A_ and A_2B_ adenosine receptors after infection. This in turn led to the release of Vascular Endothelial Growth Factor (VEGF) by the pericytes leading to pericyte detachment and endothelial cell proliferation resulting in overall BBB impairment [189].

Using rat models of meningitis, a dose-dependent increase of BBB permeability was observed after inoculation with *Hib* LPS [190]. Later studies showed that the permeability of the BBB was also increased in rats after inoculation with *H. influenzae* outer membrane vesicles (OMV), suggesting a role for these vesicles in transporting *Hib* LPS to the CSF during meningitis [191]. Furthermore, it was shown that zyxin, a cytoskeletal protein implicated in the protection of TJs in the BBB, is critical for the integrity of the BBB and, as a consequence, for protecting against invading pathogens such as *H. influenzae* [192]. Overall, the inflammatory response of patients during the infection with *H. influenzae* is determined by several virulence factors including the capsule, adhesion proteins, pili and outer membrane proteins as well as LPS and IgA1 protease [193].

## 5. Conclusions

A variety of studies have focused on how pathogens cross the blood-CNS barriers, mostly focusing on the BBB. Although significant progress has been made in identifying mechanisms of host–pathogen interactions during bacterial meningitis, additional efforts towards identifying bacterial and host cell targets are needed. The diversity of mechanisms used by these pathogens presents the need for further research. To this end, the identification of common mechanisms used by multiple pathogens will be of great significance and further assist the development of effective therapies. Of high importance is the distinction between the mechanisms the pathogens use for crossing the BCSFB and the BBB. Pathogens exploiting epitopes of both barriers would present interesting targets for the development of therapeutics.

## Figures and Tables

**Figure 1 ijms-20-05393-f001:**
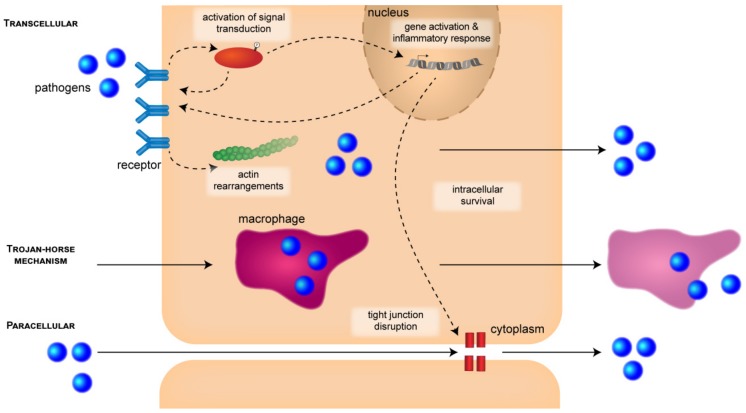
CNS entry pathways and stages during the pathogenesis of bacterial meningitis. Bacterial pathogens can cross the BBB or BCSFB paracellularly between neighboring cells, in a “Trojan horse” fashion inside infected host macrophages, or transcellularly by invading epithelial or endothelial cells. Cellular entry can be launched by the zipper mechanism involving binding to host cell receptors or by the trigger mechanism (see main text for details). During this process, activation of signal transduction pathways can cause initiation of actin rearrangements. Activation of signal transduction can also be triggered during the transcellular pathway or the “Trojan horse” mechanism, but is not indicated in the figure for reasons of clarity. Once in the cytoplasm, the pathogens need to survive inside the cells for further disease progress. Activation of host genes causes an inflammatory response that can lead to disruption of tight junctions.

**Figure 2 ijms-20-05393-f002:**
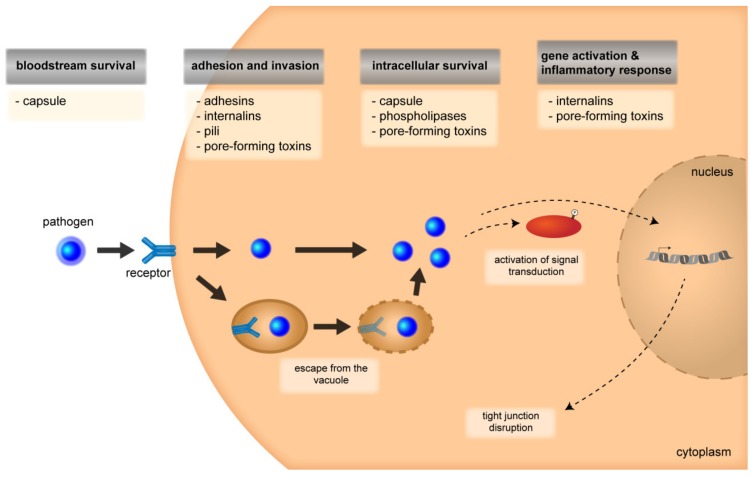
Multiple virulence factors are involved in the different steps of pathogenesis during bacterial meningitis. Expression of a capsule can support bloodstream survival of both gram-positive and gram-negative bacteria. It has been described that down-regulation of capsule expression occurs during adhesion to and invasion into host cells, which is mediated by adhesins, internalins, pili, and pore-forming toxins. Pore-forming toxins can also be involved during escape from vacuoles inside of host cells, as well as intracellular survival. These steps are further supported by pore-forming toxins and the capsule. Several virulence factors, including internalins and pore-forming toxins, activate host cell signal transduction and mediate gene activation causing an inflammatory response.

**Table 1 ijms-20-05393-t001:** Evidence for the involvement of gram-positive and gram-negative bacteria during brain entry at the BBB and BCSFB.

Pathogen	Entry Mechanisms		Major Virulence Factors	
**Gram-Positive**	**BBB**	**BCSFB**	**BBB**	**BCSFB**
*L. monocytogenes*	Transcellular route [54] “Trojan horse” mechanism within leukocytes [54] Retrograde migration within axons of cranial nerves [54]	Transcellular route [57]	Major invasion protein InlB inducing receptor-mediated endocytosis [25] Bacterial surface protein InlF interacting with surface vimentin [58] Pore-forming cytolysin LLO inducing signaling pathways (NF-κB, MAPK) [64,66] ActA promoting F-actin-based intracellular motility [68]	Major invasion proteins InlA and InlB inducing receptor-mediated endocytosis [34,57]
*S. suis*	Invasion at low rates in porcine models [73,78].	Invasion demonstrated for porcine and human in vitro models [48,72] Possibly “Trojan-horse” mechanism [85]	Enolase increasing BBB permeability [80] Suilysin inducing pore formation in membranes [81]	Regulation of capsule expression [48,72]
*S. pneumoniae*	Translocation across BBB in vivo and in vitro [87,194]	Only attachment observed in an in vivo mouse model during late stages of infection with high levels of bacteremia [87].	Altered expression of the capsule for attachment [92,93] Interaction with BBB through NadA [96] Pore-forming toxin pneumolysin [101]	
GBS	Traversal of BBB in vivo and in vitro [49,109,195]		Expression of cell-wall anchored pili [111] PilA: promoting attachment of GBS [112] PilB: mediating internalization [112]	
**Gram-Negative**	**BBB**	**BCSFB**	**BBB**	**BCSFB**
*E. coli*	Traversal of BBB in vivo and in vitro [22]	Traversal of BCSFB in vitro [121]	Attachent facilitated by type 1 fimbriae and OmpA [125,126] Invasion induced by IbeA [127] and CNF1 [128] Intracellular survival promoted by the *E. coli* K1 capsule [136]	Role of fimH during adhesion [121] Involvement of OmpA, FimH and IbeA in invasion [121]
*N. meningitidi*	Traversal of BBB in vivo and in vitro [2,196]	Traversal of BCSFB in vitro of choroid plexus epithelial cells [48] Invasion of outer BCSFB in induced pluripotent stem cell-derived brain endothelial cells [149]	Protective function of the polysaccharide capsule during bloodstream survival but attenuated tissue invasion [151] Adherence through pili and surface exposed proteins [154] Invasion is facilitated by Opa and Opc [160]	Capsule attenuates invasion in vitro [48]
*H. influenzae*	Traversal of BBB in vitro [51]	Traversal of BCSFB in vitro [47]	Entry via binding of PAFR [186,187] Attachment facilitated by binding of the laminin receptor [99]	Capsule and fimbriae attenuate invasion [47] Invasion if *H. influenzae* was observed as intracellular bacterium [47]

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
