# Peer review of "Virulence Factors of Meningitis-Causing Bacteria: Enabling Brain Entry across the Blood–Brain Barrier"

_ijms, 2019, doi:10.3390/ijms20215393_

Round 1

Reviewer 1 Report

The authors compiled a nice review on virulence factors of meningitis-causing bacteria which are involved in crossing the blood-brain barrier. They summarized the current knowledge regarding the spectrum of virulence factors, common and individual strategies  that meninigits-associated bacteria can use to interact and cross cellular barriers such as the blood-brain or blood-cerebrospinal fluid barrier. Overall, the manuscript is already quite complete and covers most of the relevant aspects of this topic. A few sentences should be rephrased, some typos should be corrected. Furthermore, in some parts of the manuscript it would also be good to describe some more relevant virulence factors to further improve and complete the review:

1, l. 13: Please rephrase "…deploy a host of different virulence factors…." 1, l. 14 and throughout the manuscript: To my knowledge the term "transverse" does not exist as a verb. This should probably read as "traverse". 1, l. 34: "blood-brain barriers" should probably read as "blood-brain barrier". 1, l. 37: "host’s cells" should probably read as "host cells". 2, l. 14: In this context I feel that reference [9] is not correct, but should rather be reference [8]. 2, l. 33: "Colonialization" should read as "colonization". 3, l. 11, 12: References 16 to 18 are correct, but quite old. Maybe also a more recent paper could be cited. 3, l. 32: Please move the empty space to the correct position: "…The" triger mechanism…". 4, 45: "cell-wall" should read as "cell wall". 4, l. 45-46: This is the only sentence in the entire manuscript where "Gram-negative" and "Gram-positive" have been written with capital letter. Please harmonize the spelling in the entire manuscript. 5, l. 6: "contain" may not be the optimal term in this context. What about "embank"? 7, l. 23: "Enolase" should probably read as "enolase". 8, section about S. pneumoniae: Here it may be good to briefly mention the role of the platelet-activating factor receptor (PAFR) and poly-immunoglobulin receptor (pIgR) for adhesion and internalization into endothelial cells. 9, section about group B streptococci: Here it may be good to also mention the interaction of BspC and vimentin as well as the role oft he hyaluronidase HylB for opening of the BBB. 9, l. 44: "Gram" should read as "gram". 10, l. 3: "ibeA" should be written in italics. 10, l. 4: "Cytotoxic" should read as "cytotoxic", "OmpA" as a gene name should be written in italics and with a small letter, "ibeA" should be written in italics. 10, l. 31-32: Please rephrase this sentence, because pathogenic E. coli do not have a brain ;-) 10, section on E. coli: This section would benefit from a short description oft he function of the macrophage migration factor (MIF) in disruption of tight junctions. 11, section on Neisseria meningitidis: Please introduce the factor H-binding protein here as well as the importance of cortical plaque formation. 11, l. 41: Please add An empty space in front of "Furthermore". 12, l. 50: "This in turn led to in the release…" should read as "This in turn led to the release…". 13-14, Table 1: Please write genus and species names as well as gene names in italics and use a unified format in all lines and columns of the table, e.g. remove the enumeration characters; "in vitro" and "in vivo" should be written in italics; when you summarize E. coli factors involved in interaction with the BCSFB, please refer to the gene products, i.e. FimH, OmpA, and IbeA.

Author Response

We thank the reviewer to state that we compiled a “nice review” which is “already quite complete” and for his helpful comments. In the following we address all comments of the reviewer.

1, l. 13: Please rephrase "…deploy a host of different virulence factors…."

Rephrased to “…use a variety of different virulence factors…”

1, l. 14 and throughout the manuscript: To my knowledge the term "transverse" does not exist as a verb. This should probably read as "traverse".

We changed the term “transverse” to “traverse” throughout the manuscript.

1, l. 34: "blood-brain barriers" should probably read as "blood-brain barrier".

Since both barriers were addressed, we changed the wording from “…crossing the blood-brain barriers…” to “…crossing into the brain…”

1, l. 37: "host’s cells" should probably read as "host cells".

Changed to “host cells”

2, l. 14: In this context I feel that reference [9] is not correct, but should rather be reference [8].

We cite now both former references [8] and [9] (now [9] and [10]).

2, l. 33: "Colonialization" should read as "colonization".

Corrected as requested.

3, l. 11, 12: References 16 to 18 are correct, but quite old. Maybe also a more recent paper could be cited.

We now additionally cite a newer reference.

3, l. 32: Please move the empty space to the correct position: "…The" triger mechanism…".

Corrected as requested.

4, 45: "cell-wall" should read as "cell wall".

Corrected as requested.

4, l. 45-46: This is the only sentence in the entire manuscript where "Gram-negative" and "Gram-positive" have been written with capital letter. Please harmonize the spelling in the entire manuscript.

We have harmonized spelling to “gram-negative” and “gram-positive”.

5, l. 6: "contain" may not be the optimal term in this context. What about "embank"?

Changed as requested.

7, l. 23: "Enolase" should probably read as "enolase".

Corrected as requested.

8, section about S. pneumoniae: Here it may be good to briefly mention the role of the platelet-activating factor receptor (PAFR) and poly-immunoglobulin receptor (pIgR) for adhesion and internalization into endothelial cells.

We now mention the roles of PAFR and pIgR.

9, section about group B streptococci: Here it may be good to also mention the interaction of BspC and vimentin as well as the role of the hyaluronidase HylB for opening of the BBB.

We now mention the interaction of BspC and vimentin and the role of HylB for opening of the BBB.

9, l. 44: "Gram" should read as "gram".

Corrected as requested.

10, l. 3: "ibeA" should be written in italics.

We write now “IbeA”, since we refer to the protein.

10, l. 4: "Cytotoxic" should read as "cytotoxic", "OmpA" as a gene name should be written in italics and with a small letter, "ibeA" should be written in italics.

Corrected as requested.

10, l. 31-32: Please rephrase this sentence, because pathogenic E. coli do not have a brain ;-)

We rephrased the sentence accordingly.

10, section on E. coli: This section would benefit from a short description of the function of the macrophage migration factor (MIF) in disruption of tight junctions.

We have now included a description of the function of MIF in TJ disruption.

11, section on Neisseria meningitidis: Please introduce the factor H-binding protein here as well as the importance of cortical plaque formation.

We have included factor H-binding protein and the importance of cortical plaque formation.

11, l. 41: Please add An empty space in front of "Furthermore".

Corrected as requested.

12, l. 50: "This in turn led to in the release…" should read as "This in turn led to the release…".

Corrected as requested.

13-14, Table 1: Please write genus and species names as well as gene names in italics and use a unified format in all lines and columns of the table, e.g. remove the enumeration characters; "in vitro" and "in vivo" should be written in italics; when you summarize E. coli factors involved in interaction with the BCSFB, please refer to the gene products, i.e. FimH, OmpA, and IbeA.

Corrected as requested.

Reviewer 2 Report

The review on virulence factors of meningitis-causing bacteria presented by the authors is updated and its distribution is adequate to achieve a compression of the subject. In addition, adding an abbreviations section makes it easier to follow the terms throughout the text.

The manuscript is novel in that it has collected the bacteria that cause meningitis

Point 2 could be extended, since in the title of the review the authors highlight “virulence factors”. For example, some proteins that facilitate the transport of pathogenic bacteria and their regulation. There is an article that reviews this

New Virulence Factors Identified in Pneumococcal Meningitis.Trends in microbiology Volume 27, Issue 11, November 2019, Pages 895-896. Brain–blood barrier breakdown and pro-inflammatory mediators in neonate rats submitted meningitis by Streptococcus pneumonia. Brain Research.Volume 1471, 30 August 2012, Pages 162-168

Regarding the description of the bacteria causing meningitis, the authors have highlighted the most interesting aspects. I agree with what was presented in this review.

Author Response

We thank the reviewer to mention that our review is “updated and its distribution is adequate to achieve a compression of the subject”, and for his helpful comments.

We now mention the information asked for by the reviewer (reference Schmidt et al. 2019 (cited in “New Virulence Factors Identified in Pneumococcal Meningitis”) and reference Barichello et al. 2012 (“Brain–blood barrier breakdown and pro-inflammatory mediators in neonate rats submitted meningitis by Streptococcus pneumoniae”)) in the revised version of our manuscript. Since both references refer to data obtained with Streptococcus pneumoniae we have included this information under point 4.1.3. (Streptococcus pneumoniae).

Reviewer 3 Report

Herold et. al, presented a review of virulence factors that enable brain entry for bacteria. The review also included information about barriers of CNS and stages in pathogenesis of bacterial meningitis etc. Article is well articulated and provides good overview of the role of virulence factors. In this regard, I recommend the article for publication.

Author Response

We thank the reviewer to state that our “article is well articulated and provides good overview of the role of virulence factors” and his positive comments.

Reviewer 4 Report

Herold et. al review "Virulence factors of meningitic-causing bacteria: enabling brain entry across the blood-brain barriers" discusses the virulence factors that mediate adhesion, invasion, survival, cell signaling activation, inflammatory response and ultimately the affect on the barriers of the BBB and BCSF.  The manuscript was informative and concisely described the cellular effects of both gram-positive and gram-negative bacteria.  Overall, I thought the latter part of the review (Section 4) was very well written and informative.  The majority of my concerns are in the first half of the review.

The introduction does not flow smoothly and highlights the same concepts several times...it would be significantly improved with a few additional highlights of the rest of the review...following your brief intro of the barriers of the CNS...transition 1st to attachement/entry then to intracellular survival, and finally disruption of the barrier integrity.

Expand on the barriers of the CNS...specific tight junctions, efflux transporters, nutrient transporters, etc. These specific properties are mentioned in section 4 but are not explained at all in sections 2.1 and 2.2.

Expand on the in vivo/vitro models of the BBB and BCSFB in section 2.1 and 2.2.  Barrier models are discussed in section 4 but with no background they are confusing.  Adding the model information in the table would also significantly strengthen the table.

The paracellular/transcellular routes are well described throughout the manuscript but the "trojan horse" model could be expanded upon further.  Also the zipper and trigger mechanisms are discussed throughout the manuscript...highlighting them in the figure may be useful.

Final major concern is that there seems to be a lot of repetition throughout sections 1-3 and with information in section 4.  The review would be significantly improved if sections 1-3 were written as more "general" information and section 4 was used as specific examples for each gram-positive/negative bacteria. 

Review appeared to be missing several references. (page-line number)

(2-2;4-7;4-31; 5-4;5-9;5-15;5-21;6-10)

Make arrows larger in Figure 1 and 2.

Figure 1 legend  explains transcellular mechanisms in activating signal transduction pathways...does the Trojan horse or paracellular routes affect these mechanisms.

Section 3.2.1 the authors mention that pathogens have to overcome cellular defense mechanisms...please expand.

Author Response

We thank the reviewer to state that “the manuscript was informative and concisely described the cellular effects of both gram-positive and gram-negative bacteria”, that the section 4 of the review was “very well written and informative”, and for his helpful comments to improve the manuscript.

The introduction does not flow smoothly and highlights the same concepts several times...it would be significantly improved with a few additional highlights of the rest of the review...following your brief intro of the barriers of the CNS...transition 1st to attachement/entry then to intracellular survival, and finally disruption of the barrier integrity.

We have now rephrased the introduction according to the reviewers’ suggestions.

Expand on the barriers of the CNS...specific tight junctions, efflux transporters, nutrient transporters, etc. These specific properties are mentioned in section 4 but are not explained at all in sections 2.1 and 2.2.

We now mention these specific properties in the sections 2.1. (Blood-Brain Barrier) and 2.2. (Blood-Cerebrospinal Fluid Barrier) of the revised version of the manuscript.

Expand on the in vivo/vitro models of the BBB and BCSFB in section 2.1 and 2.2. Barrier models are discussed in section 4 but with no background they are confusing. Adding the model information in the table would also significantly strengthen the table.

In vivo and in vitro models of the BBB and BCSFB are now mentioned in the sections 2.1 (Blood-Brain Barrier) and 2.2. (Blood-Cerebrospinal Fluid Barrier). General information regarding the distinction between in vivo and in vitro was already included in the table with references. More detailed information on the model systems was not added as we believe that the table should give a short and clear summary and more detailed information would make it rather unclear.

The paracellular/transcellular routes are well described throughout the manuscript but the "trojan horse" model could be expanded upon further. Also the zipper and trigger mechanisms are discussed throughout the manuscript...highlighting them in the figure may be useful.

We have expanded on the “Trojan horse” mechanism in section 3.1.1. (CNS Entry routes). For reasons of clarity we have not introduced the zipper and trigger mechanism into figure 1, but we now mention these mechanisms in the figure legends, with referral to the main text for more information.

Final major concern is that there seems to be a lot of repetition throughout sections 1-3 and with information in section 4. The review would be significantly improved if sections 1-3 were written as more "general" information and section 4 was used as specific examples for each gram-positive/negative bacteria.

We have tried to remove repetitive sentences from the sections 1-3 to improve the quality of the manuscript. We believe that sections 1-3 are written in a general fashion (we do not refer to specific pathogens in these sections) and provide an appropriate introduction for section 4 with examples for specific pathogens.

Review appeared to be missing several references. (page-line number)

(2-2;4-7;4-31; 5-4;5-9;5-15;5-21;6-10)

We have added references at the indicated positions.

Make arrows larger in Figure 1 and 2.

We have enlarged the arrows in figures 1 and 2.

Figure 1 legend explains transcellular mechanisms in activating signal transduction pathways...does the Trojan horse or paracellular routes affect these mechanisms.

We have introduced an appropriate sentence into the legend of figure 1.

Section 3.2.1 the authors mention that pathogens have to overcome cellular defense mechanisms...please expand.

We have expanded on the statement that pathogens have to overcome cellular defense mechanisms in section 3.2.1.